

# Evaluation of nutrient content of different harvest stages in switchgrass (*Panicum virgatum* L.) cultivars

Semih Acikbas

Department of Field Crops, Siirt Universty, Siirt, Türkiye

Corresponding author
Semih Acikbas,
semihacikbas@siirt.edu.tr

## ABSTRACT

*Panicum virgatum* L., also known as switchgrass, is a warm season short-day, C4-perennial crop gaining attention in feedstock production. The harvest stage of the forage crops and the cultivars used directly affect the feed quality as animal feed. In this study, differences were evaluated in various macro element contents of switchgrass (*Panicum virgatum* L.) cultivars at different harvest stages. Eight different switchgrass cultivars (Alamo, BoMaster, Kanlow, Cave in Rock, Long Island, Shawnee, Shelter, and Trailblazer) were evaluated in the study. According to the results of the study, significant differences were found among the cultivars studied for phosphate (P), potassium (K), calcium (Ca), and magnesium (Mg). The Long Island cultivar had the highest mineral concentration (0.223% P, 1.049% K, 0.581% Ca, and 0.201% Mg). As harvest proceeded, the levels of P and K in the plants decreased while the levels of Ca and Mg increased. The highest P (0.223%) and K (1.030%) values were obtained from plants harvested at the pre-flowering stage, while the highest Ca (0.538%) and Mg (0.183%) values were obtained from plants harvested at the full flowering stage. Switchgrass cultivars were found to be insufficient to meet animal needs for P and Ca content throughout the plant's harvest stages. Mg was found to be sufficient in all harvest stages, while K was sufficient in the first two harvest stages, pre-flowering and 50% flowering, but insufficient in the full flowering stage. The dry matter's Ca/P ratio, regardless of cultivar or harvest stage, posed no risk to animal health. The K/(Ca+Mg) ratio varied between 0.807 and 1.235 depending on the cultivar and harvest stage. This ratio does not pose any risk to animal health. Based on these results, it would be appropriate to supplement feed rations with materials containing P and Ca or to use switchgrass in animal feeding together with feeds rich in these minerals when using dry matter obtained from switchgrass cultivars alone as roughage. In general, although *Panicum virgatum* has an important potential as a feed source in animal nutrition, it needs to be supplemented with nutrient-containing additives. It would also be appropriate to determine the nutrient content of these varieties under different ecological conditions.

## INTRODUCTION

Switchgrass (*Panicum virgatum* L.) is a perennial warm-season C4 grass, native to North America, primarily utilized as a biofuel crop due to its high biomass production (*Yuan et al., 2015*; *Aurangzaib et al., 2018*; *Niu et al., 2022*). Its notable features include its wide adaptability, significant tolerance to various abiotic stresses including cold and drought, and its ability to increase soil organic carbon and biodiversity (*Liebig et al., 2008*; *Robertson et al., 2011*; *Balsamo et al., 2015*). Switchgrass can be utilized as forage for ruminant animals through grazing, as well as for silage and hay as roughage (*Zhao et al., 2017*; *Taranenko et al., 2019*; *Eliş & Özyazıcı, 2019*).

The maturity stage, harvest time, and cultivars used significantly affect the quality of switchgrass as animal feed (*Mohammed et al., 2015*). It is known that the nutrient content of feeds to be used in the feeding of ruminant animals as roughage varies according to the harvesting stages of plants. Additionally, the chemical compositions of feeds vary significantly depending on factors such as plant species, soil structure, fertilization, vegetation period, and climate (*Markovic et al., 2014*). Early harvesting of plants boosts feed quality but decreases product quantity, whereas later harvesting increases biological yield but reduces feed quality and palatability due to lignification (*Gürsoy & Macit, 2020*). Studies conducted in some regions of the United States (Pennsylvania, Wisconsin, and Illinois) on switchgrass have reported that delaying the harvest from mid-fall to spring resulted in a reduction of macronutrients (*Adler et al., 2006*; *Serapiglia et al., 2016*). Additionally, due to the lower uptake of macronutrients resulting from the delayed harvest, less nutrient supplementation was required for the growth of the following year (*Vogel et al., 2002*).

Minerals are essential for animal life and understanding the mineral balance and nutritional content of roughages used in animal feed is crucial for animal health. Animals' mineral needs vary based on species, breed, age, gender, growth, health, pregnancy, and milk output, as well as the quantity and bioavailability of minerals consumed (*Pinotti et al., 2021*). Switchgrass provides a sustainable advantage in ruminant nutrition due to its perennial nature and low input cost (*Giannoulis et al., 2017*). According to *Gülümser et al. (2017)*, macro elements including phosphorus (P), potassium (K), calcium (Ca), and magnesium (Mg) are just as important in animal nutrition as other nutrients. In cases of mineral deficiency, serious health issues such as loss of appetite, decreased productivity, skin diseases, digestive system disorders, and bone abnormalities can occur. If necessary precautions are not taken, these conditions can even lead to death (*Ensminger, Oldfield & Heinemann, 1990*; *Mccaughan, 1992*; *Underwood & Suttle, 1999*; *Schonewille, 2013*; *Swerczek, 2018*).

The analysis and determination of mineral concentrations in feeds is crucial for appropriate mineral supplementation in animal diets when necessary. To achieve maximum and high-quality animal products, it is vital to understand the differences in nutrient content between harvest stages of plants evaluated as feed crops. Therefore, the purpose of this study was to evaluate the variations in macroelement content between harvest stages and cultivars of switchgrass (*P. virgatum* L.) and their effect on feed quality.

## MATERIALS AND METHODS

### Plant materials and growth conditions

The study was conducted under semi-arid climate conditions in a field in Siirt province, in the Southeastern Anatolia Region of Turkey (37°58′13.20″N–41°50′43.80″E), in 2019 and 2020. The plant material of the study consists of eight switchgrass cultivars. Alamo, BoMaster, and Kanlow are lowland ecotypes, Cave in Rock, Long Island, Shawnee, and Shelter are upland types, while Trailblazer is intermediate.

The climate data of the study were recorded at Siirt Airport, located 1 km from the trial site, and the data was obtained from the Turkish State Meteorological Service. The study area, located in the Southeastern Anatolia Region of Turkey, is generally dominated by a continental climate, with all four seasons experienced in their most distinct forms. Summers are hot and dry, while winters are harsh. Rainfall is scarce between June and October. Most of the precipitation occurs in the spring, autumn, and winter months. According to the long-term (1939–2020) meteorological data (*Turkish State Meterological Service, 2020*) for Siirt province, the current climate in the region is semi-arid. Table 1 shows the average temperature and total precipitation for the studied years, as well as long-term averages. The annual average temperature readings for cultivation years 2019 and 2020 were close to one another and somewhat higher than the long-term average. The first year of the trial had higher total annual precipitation than the second year and long-term values, indicating that it was wetter (Table 1).

In the study, the switchgrass plants used were established in 2015, and the research was conducted in the 4th and 5th years of the plants. The study was conducted with four replications, 20 cm row spacing, ten rows, and 400 seeds per square meter. The main plot size was 2 m × 4.5 m, and through sub-parceling, the plot size for each cutting period was considered as 2 m × 1.5 m. In the first year of the trial, triple superphosphate (% 42 $P_2O_5$) was applied to each plot at a rate of 12 kg $P_2O_5$ per hectare in early spring, followed by urea (% 46 N) at a rate of 9 kg/ha of pure nitrogen in May. The switchgrass experimental field was properly irrigated each year using a drip irrigation system; weeds were controlled with hand hoeing when observed, and the annual maintenance of the field was carried out regularly. The subject of the study comprised eight different switchgrass cultivars and three different harvest stages (I. Pre-flowering stage, II. 50% flowering stage, and III. Full flowering stage). Due to the presence of cultivars with different ecotypes, harvesting was carried out at the appropriate harvest stages for each cultivar in each year respectively. In general, upland ecotype varieties (Cave in Rock, Long Island, and Shawnee) reached the cutting stages earlier, while lowland ecotype varieties (Alamo, BoMaster, Kanlow) reached the cutting stages later. The Trailblazer cultivar is classified as intermediate and showed growth patterns between these ecotypes. Throughout the study, the lowland ecotype varieties were harvested in the pre-flowering stage on July 8–21 and July 10–19, in the 50% flowering stage on July 24–28 and July 23–26, and the full flowering stage on August 1–2 and July 30–August 2 in 2019 and 2020, respectively. Similalry, for the upland ecotype varieties, these dates were June 15–24 and June 13–18 for the pre-flowering stage, June 24–July 5 and June 23–27 for the 50% flowering stage, and July 8–16 and July 8–12 for the

**Table 1 Average temperature and total precipitation for long-term (1939–2020) and study years (2019 and 2020).**

| Months | Average temperature (°C) | | | Total precipitation (mm) | | |
|---|---|---|---|---|---|---|
| | **2019** | **2020** | **LTA (1939–2020)** | **2019** | **2020** | **LTA (1939–2020)** |
| January | 4.0 | 3.5 | 2.7 | 94.0 | 63.8 | 95.1 |
| February | 5.7 | 3.7 | 4.3 | 110.4 | 137.2 | 98.4 |
| March | 8.3 | 11.1 | 8.4 | 185.2 | 229.6 | 112.6 |
| April | 11.9 | 14.1 | 13.8 | 166.6 | 158.6 | 105.6 |
| May | 21.9 | 20.6 | 19.4 | 63.0 | 40.4 | 63.5 |
| June | 29.1 | 27.2 | 26.0 | 1.2 | 0.2 | 9.2 |
| July | 30.5 | 31.6 | 30.6 | 0.0 | 5.2 | 2.7 |
| August | 31.8 | 30.6 | 30.3 | 40.4 | 3.8 | 1.7 |
| September | 26.4 | 29.0 | 25.5 | 51.4 | 0.0 | 6.9 |
| October | 20.9 | 21.7 | 18.3 | 75.8 | 0.0 | 50.3 |
| November | 11.9 | 11.9 | 10.5 | 51.4 | 57.8 | 81.9 |
| December | 7.5 | 6.5 | 4.8 | 75.8 | 38.6 | 94.9 |
| X – Σ | 17.5 | 17.6 | 16.2 | 915.2 | 735.2 | 722.8 |

**Note:**
LTA, long-term average; X, mean; Σ, total.

full flowering stage. For the Trailblazer cultivar, the pre-flowering stage was on July 2 and June 25, the 50% flowering stage on July 10 and July 9, and the full flowering stage on July 21–23, with harvesting done according to the cutting period.

Each year, soil samples were collected from the study area in October, from a depth of 0–20 cm, by combining samples from five different locations into a single composite sample to determine some physical and chemical properties. Soil samples were analyzed for texture, pH, electrical conductivity (EC), lime, organic matter, available phosphorus, and potassium contents at Science and Technology Application and Research Center of Siirt University. The pH and EC were determined using the 1:2.5 (w/v) soil-water mixture method (*Horneck et al., 1989*). Lime was detected using the calcimeter method. The Bouyoucus hydrometer method was used for texture analysis (*Bouyoucos, 1951*). Soil organic matter was determined using the Walkley Black wet combustion method (*Nelson & Sommers, 1982*). Available phosphorus was calculated by sodium bicarbonate method using a spectrophotometer (*Olsen et al., 1954*), and available potassium was determined using the ammonium acetate method with a flame photometer (*Ferrando et al., 2020*). Switchgrass is grown in clay-textured soils that are non-saline, mildly alkaline, and moderately calcareous, with low amounts of organic matter and accessible phosphorus, but high available potassium. The extractable calcium and magnesium concentration is confirmed to be adequate (Table 2).

## Macronutrient analyses in forage samples

The study consisted of 32 plots in total, with eight varieties and four replications, and samples were taken from the same plots each year. From the green forage harvested from

**Table 2 Some physical and chemical properties of the research area soils (0–20 cm)[*].**

| Soil property | Unit | Value | |
|---|---|---|---|
| | | 2019 experimental site | 2020 experimental site |
| Sand | % | 24.00 | 28.35 |
| Clay | % | 58.00 | 55.65 |
| Silt | % | 18.00 | 16.00 |
| pH | | 7.95 | 7.98 |
| Electrical conductivity | dS m$^{-1}$ | 0.358 | 0.341 |
| Lime (CaCO$_3$) | % | 10.5 | 12.7 |
| Organic matter | % | 1.35 | 1.25 |
| Available P | kg P$_2$O$_5$ ha$^{-1}$ | 36 | 33 |
| Available K | kg K$_2$O ha$^{-1}$ | 1,170 | 1,140 |
| Available Ca | ppm | 29,123 | 28,145 |
| Available Mg | ppm | 1,159 | 1,255 |

Note:
[*] Analyses were conducted at the Soil Analysis Laboratory of Siirt University, Science and Technology Application and Research Center.

each plot, 500-g samples were randomly taken. The samples were further dried in an oven at 70 °C for 48 h, pulverized, and prepared for analysis. Total P, K, Ca, and Mg concentrations in the ground hay samples were determined using an IC-0904FE calibration set and a Near Infrared Reflectance Spectroscopy (NIRS) device (*Brogna et al., 2009*). Additionally, the Ca/P and K/(Ca+Mg) ratios in the samples were calculated.

## Statistical analysis

The study was conducted as a split-plot design, with cultivars assigned to the main plots and harvest stages assigned to the sub-plots. All data obtained from the study were subjected to analysis of variance according to a split-plot design with four replications. The 2-year data obtained from the study were subjected to Levene's test for homogeneity, and the data determined to be homogeneous underwent a combined analysis of variance. Differences between groups were determined using the Tukeys HSD multiple comparison test based on the F-test results (*Açıkgöz & Açıkgöz, 2001*). The relationships between nutrient elements and harvest stages across different years was determined using Pearson's correlation with Minitab® 21.4.1 software.

## RESULTS

### Analysis of variance experimental data

Analysis of variance results for the determined P, K, Ca, Mg, Ca/P, and K/(Ca+Mg) values according to the harvest stages in the studied switchgrass cultivars are presented in Table 3. The difference between years was found to be statistically significant ($p < 0.01$) for K, Ca, Mg, Ca/P, and K/(Ca+Mg), but not for P. According to the 2-year average results, statistically significant differences were obtained among cultivars ($p < 0.01$) for P, K, Ca, Mg, and Ca/P, and at the $p < 0.05$ level for K/(Ca+Mg). At the harvest stage, Mg showed statistically significant differences ($p < 0.05$), whereas other metrics were significant at the

**Table 3 Analysis of variance for years and treatments on investigated characteristics.**

| Traits/Factors | TUKEY value/F probability | | | | | | |
|---|---|---|---|---|---|---|---|
| | Year (Y) | Cultivars (C) | Harvest stage (HS) | Y × C | Y × HS | C × HS | Y × C × HS |
| Phosphorus (P) | 0.004/ns | 0.012/** | 0.004/** | 0.019/** | 0.007/** | 0.019/** | 0.030/ns |
| Potassium (K) | 0.049/** | 0.154/** | 0.063/** | 0.249/* | 0.109/** | 0.281/** | 0.436/ns |
| Calcium (Ca) | 0.015/** | 0.046/** | 0.021/** | 0.075/ns | 0.037/* | 0.095/** | 0.148/* |
| Magnesium (Mg) | 0.004/** | 0.013/** | 0.005/* | 0.020/* | 0.018/ns | 0.023/ns | 0.036/ns |
| Ca/P | 0.084/** | 0.265/** | 0.117/** | 0.429/ns | 0.203/** | 0.522/** | 0.809/ns |
| K/Ca+Mg | 0.017/** | 0.341/* | 0.217/** | 0.871/ns | 0.373/ns | 0.963/ns | 1.496/ns |

Notes:
  * $p < 0.05$.
  ** $p < 0.01$.
  ns, no significant difference.

$p < 0.01$ level. When evaluating P and K concentrations for pairwise interactions, they were found to be statistically significant. However, for Ca, the interaction between years and cultivars (Y × C) was significant at the $p < 0.05$ level, while the interactions between years and harvest stages (Y × HS) and between cultivars and harvest stages (C × HS) were statistically insignificant. Regarding the Ca/P ratio, the interaction between years and cultivars (Y × C) was statistically insignificant, whereas the interactions between years and harvest stages (Y × HS) and between cultivars and harvest stages (C × HS) were significant at the $p < 0.01$ level. In the triple interaction (Y × C × HS), only Ca concentration was statistically significant at the $p < 0.05$ level, whereas all other parameters were determined to be statistically insignificant (Table 3).

## Effect of growing seasons on macro element content

The experiment showed significant variations across the years. According to the average results of 2 years, the highest levels of K, Ca, and Mg contents were observed in plants harvested in the second year (1.048%, 0.533%, and 0.189% respectively). However, there was no statistically significant difference in terms of P content between the years for both cultivars and harvest stages. The Ca/P and K(Ca+Mg) ratios were determined as 2.340 and 1.247, respectively, for the plants collected in their first year (Fig. 1).

## Mineral contents variations among cultivars

The Long Island cultivar had the highest average P, K, Ca, and Mg contents throughout the 2-year and harvest stages, with values of 0.223%, 1.049%, 0.581%, and 0.201%. The Long Island also had a P content that was statistically comparable to the Shelter, Shawnee, and Trailblazer cultivars. Overall, the P content was prominent in lowland ecotype cultivars. The lowest values for P content were observed in the Kanlow (0.199%) and Alamo (0.202%). The cultivars with the lowest K content were Kanlow (0.687%) and Alamo (0.690%). The Kanlow showed the lowest magnesium content over 2 years and harvest stages, at 0.169% (Fig. 2). When examining the Ca/P ratio, the highest value of 2.624 was obtained in the Long Island cultivar. The Alamo, Kanlow, BoMaster, Trailblazer, and Cave in Rock were found to be statistically similar to the Long Island. The lowest values for the

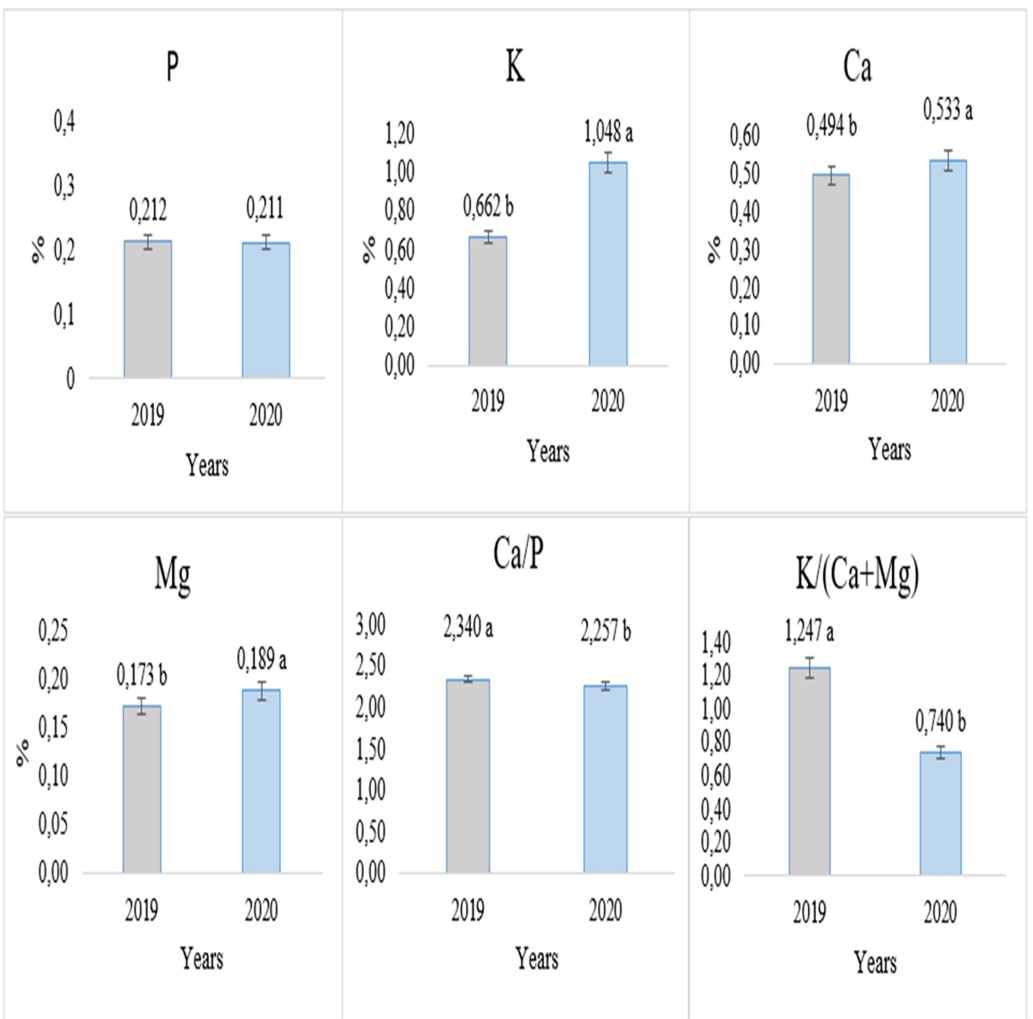

**Figure 1 Alteration of investigated parameters depending on the experimental years.**

Ca/P ratio were determined in the Shelter (2.341) and Shawnee (2.338). The Alamo and Cave in Rock had the highest K/(Ca+Mg) ratios (1.233 and 1.235, respectively), while Shelter, Shawnee, and Long Island had the lowest ratios (Fig. 2).

## Mineral content variations among harvest stages

According to the average results of the 2 years, when assessing the contents of P and K, significant declines in the specified mineral contents were observed as the harvest stage progressed. The highest P and K contents were obtained during the pre-flowering stage, while the lowest values were obtained during the full flowering stage. The Ca and Mg contents increased throughout the harvest stage, peaking during full bloom. Similarly, the Ca/P and K/(Ca+Mg) ratios increased as the harvest stage progressed. The highest values were obtained during the full flowering stage (Fig. 3).

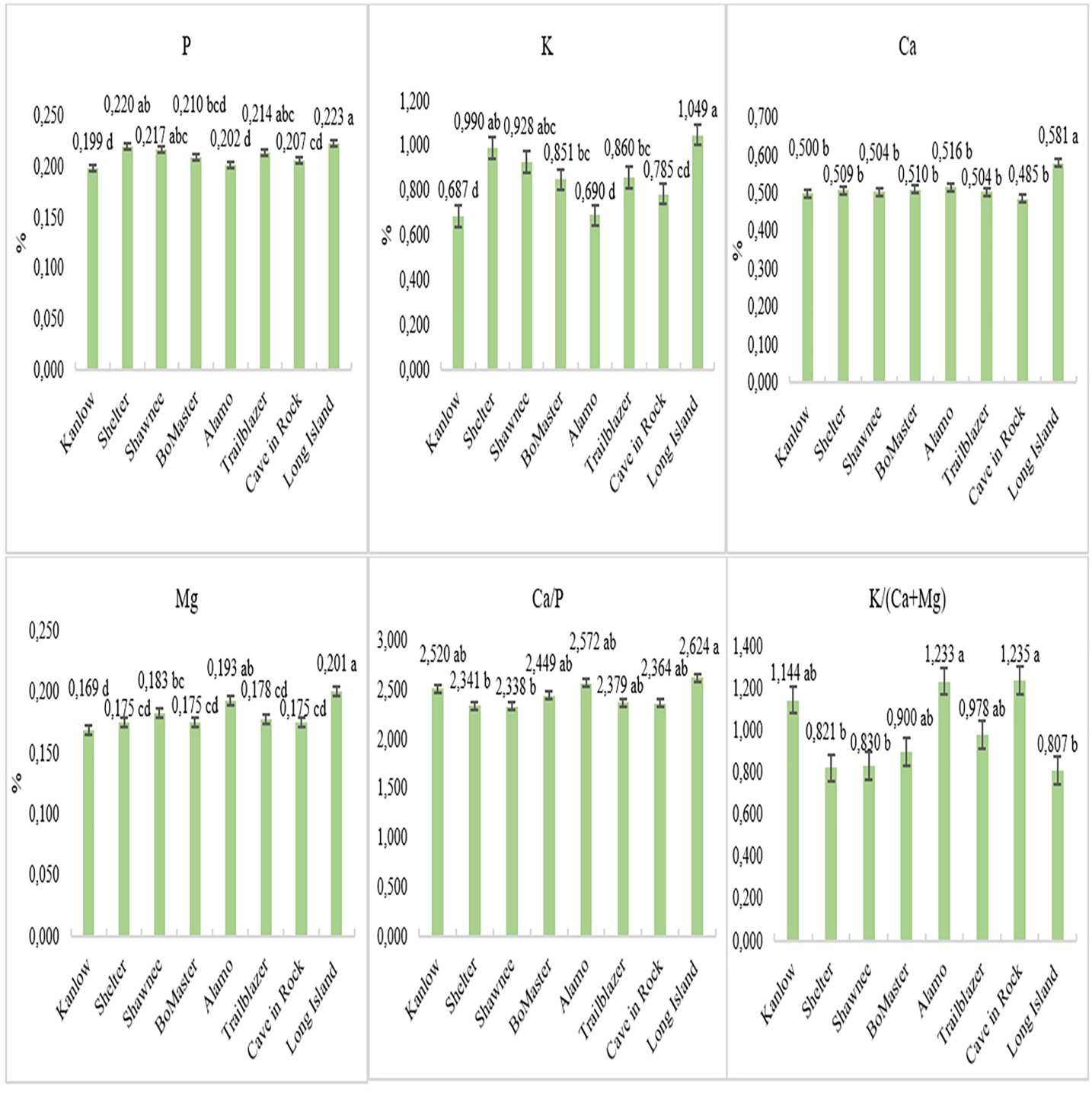

**Figure 2 Mineral matter content of switchgrass cultivars in both years.**

## Interaction effects of experimental factors on statistically significant traits

In the Year × Harvest stage interactions, the highest level of P was obtained during the pre-flowering stage in the first year, and it peaked again in the second year. The highest

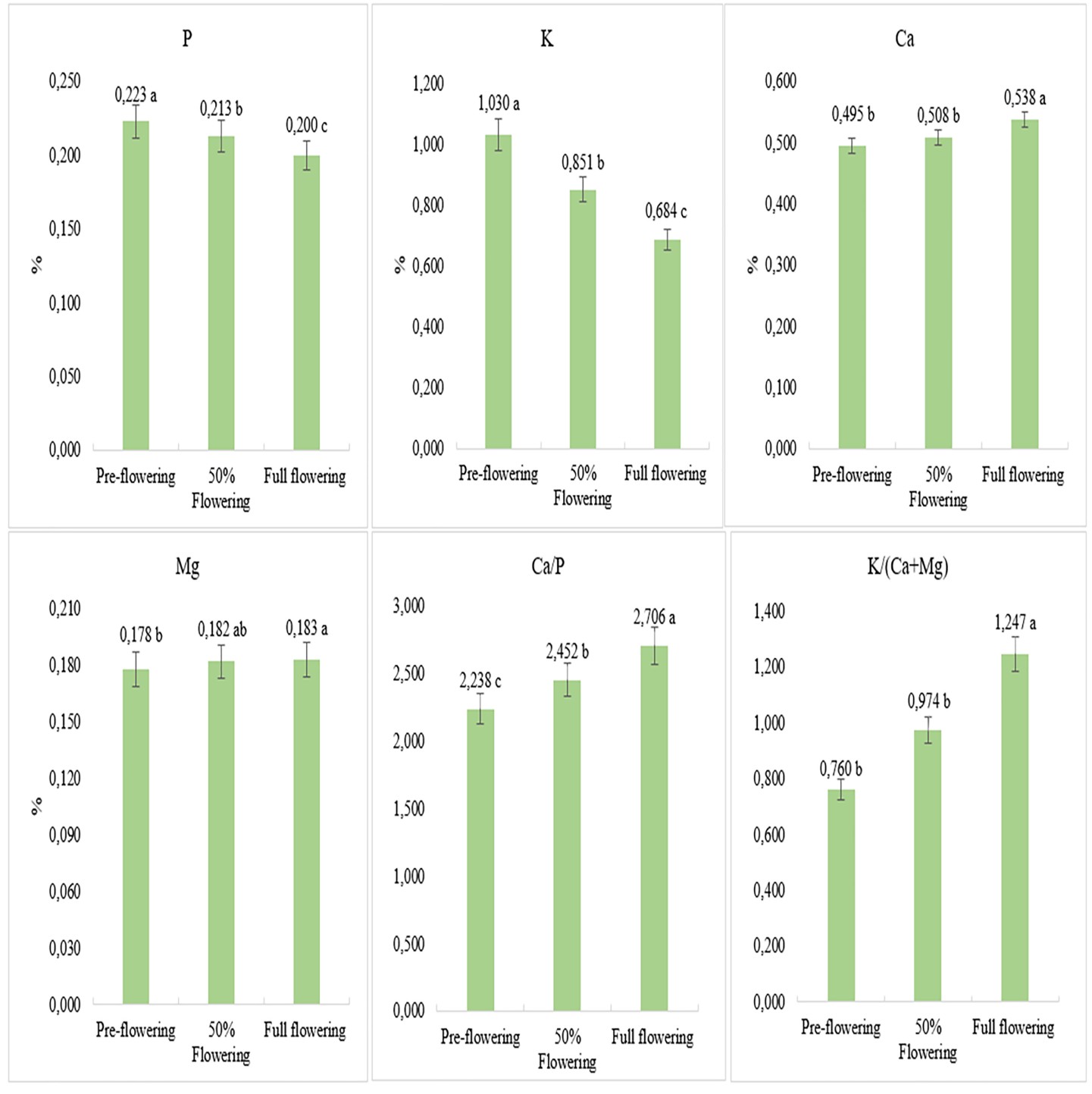

**Figure 3 Mineral matter content of switchgrass cultivars to harvest stages in both years.**

values for K were obtained during the pre-flowering stage in the second year, indicating that overall values were higher in the second year. The highest Ca values were obtained during the full blooming stage of the second year. In general, the values for Ca in the second year were higher compared to the first year. Year × Cultivar × Harvest stages

interaction for Ca were found to be statistically significant. Some cultivars showed fluctuating trends across years and did not exhibit the expected increase with the progression of the harvest stage, leading to the significance of the triple interaction in terms of Ca, Ca/P, and K/(Mg+Ca) ratios (Tables S1 and S2). The correlation analysis showing the relationships between nutrient elements and harvest stages (Year × Cultivar × Harvest stages) in different years is evaluated with Pearson's correlation analysis. According to the results of correlation analysis; there was a positive correlation between varieties and year for calcium magnesium, phosphorus and potassium. Accordingly, there was a general increase in nutrient accumulation as the growth stage progressed, except for P mineral. There was also a positive correlation between harvest times (Fig. 4).

When Year × Cultivar interaction was evaluated in the study, cultivars showed similar performances across years in terms of P, Mg, and Ca/P concentrations. However, for K and Ca, cultivars were found to have higher values in the second year. For the K/(Mg+Ca) ratio, the ratios of cultivars reached higher values in the first year. In the interaction between cultivar and harvest stages, the maximum values were attained in the pre-flowering stage for P and K concentrations, and during the full flowering stage for Ca, Ca/P, and K/(Mg +Ca) ratios (Tables S3 and S4).

## DISCUSSION

Forage crop breeding is a continuous process that has a direct effect on animal feeding quality and human nutrition as a protein source. One of the major issues in the developing world is protein malnutrition. Increasing costs in animal feeding are one of the major drivers of the price increase in protein sources (Henchion et al., 2017; Zira et al., 2023; Nadathur et al., 2024). Here, Switchgrass' nutritional quality values were evaluated in terms of harvest stages and seasons. Significant differences were obtained in cultivars and harvest stages. The study revealed that the cultivation years had varying effects on the examined nutrient elements. Although the average temperature values were generally similar, this was not the case for total precipitation. The difference between years is thought to be influenced by variations in precipitation levels over time. According to Roche et al. (2009), plant uptake of mineral compounds depends on various parameters, and meteorological conditions such as temperature, duration of sunlight, and precipitation, notably soil and air temperature, influence the concentration of mineral substances in plants. Even though switchgrass can redistribute some nutrients from shoots to roots during each growing season, significant amounts of nutrient elements are taken up with harvested biomass (Yang et al., 2009). It has been discovered that the plant mineral composition is determined by the movement of minerals from the rhizosphere to the root-shoot connection and aboveground tissues, which are substantially impacted by environmental and genotypic factors (Palmer et al., 2014).

Knowledge of the mineral content of roughage, which is a key input in animal production, is of great importance for balanced animal nutrition. According to our results, significant differences were observed in the mineral content of switchgrass cultivars. This situation may be ascribed to genetic variances among cultivars, and ecological variation affects the potential uptake of plant nutrients from the soil. In our previous studies

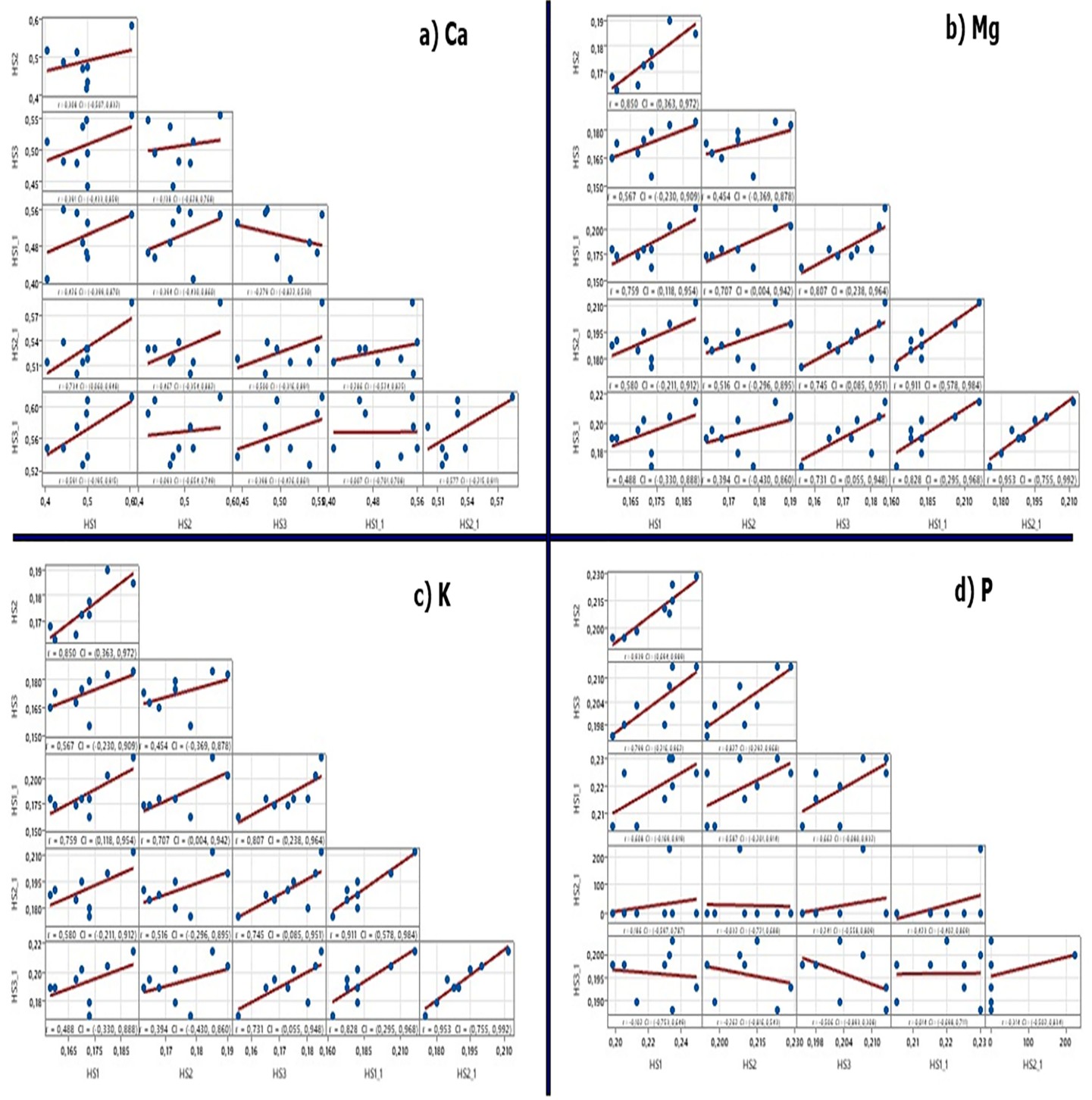

**Figure 4 The correlation of nutrient elements according to harvest stages in different years.**

conducted with different plant species in different ecosystems (*Ozyazici & Acikbas, 2019*; *Özyazıcı & Açıkbaş, 2020, 2023*) and in numerous other reports (*Lema, Cebert & Sapra, 2004*; *Markovic et al., 2014*; *Schlegel et al., 2016*), these variations have been highlighted. In

switchgrass, stems and leaves regenerate annually and live for one year, while perennial tissues such as crowns, rhizomes, and older roots live for much longer periods. Crown and rhizome tissues connect the root and shoot systems, emphasizing the significance of minerals transmitted from annual or perennial tissues. Mineral uptake and recycling form the basis of sustainable biomass production in switchgrass and other perennial herbaceous plants (*Palmer et al., 2014*).

Plants can only absorb certain nutrients within specific pH ranges (*Tkaczyk et al., 2018*). The pH range of soils supports the intensive development of soil microorganisms that effectively mineralize organic matter and increase the availability of essential mineral nutrients (*Jarociński, 2005*). When soil pH is <6.0, forms of Mg, Ca, and P become less available for plant roots, and when pH falls below 5.5, the uptake of nitrogen (N), potassium (K), and sulfur (S) becomes more difficult (*Tkaczyk et al., 2018*). Generally, plants have the best availability of macro nutrients (P, K, Ca, Mg) in the pH range of 6–7. Maintaining the correct pH range in the soil ensures the balanced absorption of nutrient elements by plants and positively affects plant growth. The pH values of the soil during the study were 7.95 and 7.98, respectively, and it is thought that this may have an effect on the uptake of the studied nutrients.

The differences in the P, K, Ca, and Mg contents of the cultivars studied under optimal fertilization conditions can be explained by the influence of the soil's other physical and chemical fractions as well as climatic conditions such as temperature, sunlight duration, and precipitation on the concentration of minerals in the plant. Since minerals cannot be synthesized within the organism, they are essential, meaning they are necessary for the continuation of vital functions. Therefore, a significant portion of these substances needed by animals is met through feed (*Eğritaş & Önal, 2015*). Plants, on the other hand, can take up the required nutrients from the soil as long as they are in an available form.

It has been reported that the P content of silage maize (*Zea mays* L.), a forage crop from the Poaceae family similar to switchgrass, ranges from 0.11% to 0.35%. Its K content ranges from 0.072% to 1.920%, Ca content from 0.17% to 0.52%, and Mg content from 0.16% to 0.34% (*Öner et al., 2011*; *Özata, Öz & Kapar, 2012*). Maize and switchgrass exhibit similar P, K, Ca, and Mg content levels.

Various references for the minimum threshold values required for P, K, Ca, Mg, Ca/P, and K/(Ca+Mg) in animal feed rations are listed in Table 4. According to *Kidambi, Matches & Gricgs (1989)*, except for the Kanlow, Alamo, and Cave in Rock cultivars, all other cultivars meet or surpass the threshold value for phosphorus (P) based on 2-year averages. However, based on the threshold value indicated by *Muller (2009)*, it is understood that none of the cultivars meet the needs of ruminants. The low concentration of P in the plant is thought to be due to the effective retention of phosphorus in the soil. According to our previous research (*Ozyazici & Acikbas, 2019*), the high extractable Ca and Mg concentrations in the research soils hinder plant uptake of phosphorus. When the cultivars were evaluated for K, the values ranged from 0.687 to 1.049, which are higher than the values reported by *Tajeda et al. (1985)* and *Kidambi, Matches & Gricgs (1989)*. Switchgrass cultivars can meet the K needs of ruminant animals in feed rations. While the Ca content of the cultivars ranged from 0.485 to 0.581, these values were higher than the

**Table 4 Threshold values (%) of some macro elements and ratios in ruminant animal feed rations.**

| P | K | Ca | Mg | Ca/P | K/(Ca+Mg) | References |
|---|---|---|---|---|---|---|
|  | 0.80 | 0.30 |  |  |  | *Tajeda et al. (1985)* |
| 0.21 | 0.65 | 0.31 | 0.10 |  |  | *Kidambi, Matches & Gricgs (1989)* |
| 0.40 |  | 0.90 |  |  |  | *Muller (2009)* |
|  |  |  |  | 1:1–2:1 |  | *National Academy of Sciences (1984)*, *Açıkgöz (2001)* |
|  |  |  |  |  | ≤2.2 | *Kidambi, Matches & Gricgs (1989)*, *Crawford et al. (1998)* |

threshold values reported by *Tajeda et al. (1985)* and *Kidambi, Matches & Gricgs (1989)*, indicating a good level of calcium. However, it was observed that all values are below the threshold value reported by *Muller (2009)*. The magnesium (Mg) content of the cultivars was likewise above the threshold value specified for addressing the demands of animals, demonstrating that the cultivars provide an appropriate quantity of magnesium for ruminant feeding (Table 4).

In animal nutrition, the mineral content of the feed is important, but the ratio between certain nutrients is also significant. The ratios between certain nutrients, such as the Ca/P and K/(Ca+Mg), are crucial for the metabolic activities and health of animals (*Kumar & Soni, 2014*). According to the threshold values (Table 4), the Ca/P ratio is generally recommended to be between 1:1 and 2:1. In our study, it was observed that the Ca/P ratio varies between 2.338 and 2.624 across the years and cultivars examined. The Ca/P ratios obtained in our study exceed the desired level. It has been reported that when this value falls below the recommended threshold, it can cause milk fever (hypocalcemia) in animals (*Açıkgöz, 2001*). When the K/(Ca+Mg) ratio is examined, the ratios of cultivars ranged from 0.807 to 1.235 over the years (Fig. 2). It was reported that the K/(Ca+Mg) ratio in forage crops should be less than 2.2 (*Kidambi, Matches & Gricgs, 1989*); some researchers have reported that if this ratio is 2.2 or higher, there is an increased risk of grass tetany (*Kumssa et al., 2020*; *Loudon et al., 2021*; *Yılmaz, 2022*). Sheep and cattle during lactation are more sensitive to grass tetany due to their elevated magnesium (Mg+2) (*Martens et al., 2018*; *McAllister et al., 2020*; *Pinotti et al., 2021*). The study's findings indicate that the observed ratios are below the stated threshold value, posing no risk of grass tetany.

The stage of plant growth/harvest time is among the most important parameters affecting the quality of forage obtained from fodder crops. In this regard, plant growth stage/harvest time is one of the most important parameters affecting the quality of forage. When the average of years for the harvest stages was evaluated, the P and K concentrations reached the highest values during the pre-flowering stage, and with the progression of the harvest stage due to increasing plant development, there was a statistically significant decrease in P and K concentrations. Similar results to those of the current study have also been reported in studies conducted on Switchgrass, where P and K, along with Cl and S minerals, were found to be much greater in late-season (post-senescence) harvests than in early-season and summer harvests (*Reynolds, Walker & Kirchner, 2000*; *Vogel et al., 2002*; *Dien et al., 2006*; *Lemus, Parrish & Wolf, 2009*; *Yang et al., 2009*). Research on other cereal

forage crops has likewise indicated a general decline in the mineral content of forage dry matter with the progression of the development stage/delayed harvest time (*Brink et al., 2006*; *Türk, Albayrak & Yüksel, 2007*; *Nordheim-Viken, Volden & Jørgensen, 2009*; *Schlegel et al., 2016*; *Özyazıcı & Açıkbaş, 2020*).

In contrast to P and K, it has been shown that the concentrations of Ca and Mg increase as the harvest stage continues, reaching their maximum values at full flowering stage. The most abundant minerals in Switchgrass tissues are K, Ca, and P (*Monti et al., 2008*; *El-Nashaar et al., 2009*). While many minerals can be repositioned from aboveground biomass during the aging process, some minerals, such as Ca and Mg, cannot be repositioned (*Dien et al., 2006*; *Lemus, Parrish & Wolf, 2009*; *Yang et al., 2009*). There was a parallel increase in Ca/P and K/(Ca+Mg) ratios as the harvest stage progressed. A comparable study conducted with populations of *Dactylis glomerata* L. matches our data, reporting a rise in the K/(Ca+Mg) ratio as the harvest stages advance (early vegetative stage, stem elongation stage, full blooming stage) (*Can & Ayan, 2017*). Using *Muller*'s *(2009)* threshold values, it was found that Switchgrass cultivars were insufficient in meeting the needs of animals in terms of P and Ca across all harvest stages. However, Mg was found to be at a sufficient level in all developmental stages, while K was considered sufficient in the first two harvest stages (pre-flowering and 50% flowering), according to the threshold value indicated by *Tajeda et al. (1985)*, but it decreased and became insufficient during the full flowering stage.

It has been determined that as the harvest time is delayed, the mineral content of the plants (except for Ca and Mg) decreases, and the highest concentration of P and K is obtained from plants harvested at the full flowering stage. However, despite the optimal level of plant nutrients in the soil, P and Ca elements, that may become unavailable due to soil and climate factors, may not be adequately taken up by the plants.

## CONCLUSION

This study, which assessed the amounts of key macro plant nutrients in different harvest stages of switchgrass cultivars, found consistent insufficiency in phosphorus (P) and calcium (Ca) throughout all phases. These data suggest that the roughage from the investigated switchgrass cultivars fails to meet the nutritional requirements of ruminants in terms of these critical macro elements. As a result, the plants' growing environment did not provide adequate phosphate and calcium. Although, the study concluded that the examined switchgrass cultivars posed no feeding concern regarding magnesium (Mg), potassium (K) was found to be adequate for animal needs during the early stages of development but became insufficient during later stages, particularly evident during the full flowering stage. If used as animal feed, supplementation of feed rations with materials containing P and Ca and/or the use of feeds rich in these minerals is recommended to ensure optimal productivity. In general, breeding efforts or soil amendment practices could positively impact nutrient uptake in switchgrass, and different management practices (*e.g.*, adjusting harvest stage) will contribute to nutrient retention in the plants.

### Funding
The authors received no funding for this work.

### Competing Interests
The authors declare that they have no competing interests.

### Author Contributions
- Semih Acikbas conceived and designed the experiments, performed the experiments, analyzed the data, prepared figures and/or tables, authored or reviewed drafts of the article, and approved the final draft.

### Data Availability
The raw measurements are available in the Supplemental File.

### Supplemental Information
Supplemental information for this article can be found online at http://dx.doi.org/10.7717/peerj.18570#supplemental-information.

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
