# Peer review of "Evaluation of nutrient content of different harvest stages in switchgrass (Panicum virgatum L.) cultivars"

_PeerJ, doi:10.7717/peerj.18570_

## Round 0.1 · original submission · Major Revisions

Please revise this manuscript in the light of comments raised by reviewers.

·

Basic reporting

The manuscript entitled “Evaluation of nutrient content of different harvest stages in switchgrass (Panicum virgatum L.) cultivars” provides an overview of the macro-elements in eight Spp. of switchgrass at three different harvest stages. The study's key findings suggest that key macro plant nutrients in different harvest stages of switchgrass cultivars, found consistent insufficiency in phosphorus (P) and calcium (Ca) throughout all phases. The authors hypothesise that to achieve maximum and high-quality animal products, it is vital to understand the differences in nutrient content between harvest stages of plants evaluated as feed crops. However, several critical aspects of the manuscript need improvement.
Abstract: There should be clear background information in the abstract that why authors are studying the nutrient content of switchgrass? Please add brief information regarding methods that help to understand the main idea of work. Clearly state wether the spp. under investigation belongs to Long Island? Clearly delineate the exact contributions and potential applications of the study to provide comprehensive understanding of the findings.
Keywords: The keywords should be revised to include more specific and novel keywords that have broader meanings and are not already part of the manuscript title. Scientific Background and Literature Review:
The scientific background is insufficiently explained. The manuscript should provide a more comprehensive overview of the background that why authors have conducted studied the switchgrass Spp. nutrients at different stages work with recent literature. It is essential to review the previous literature (previous ten years) carefully to identify actual research gaps and justify the novelty of your study. Ensure that the introduction conceptualises your research within the broader field and clearly highlight the specific questions, hypotheses and research gap your study aims to address. Please add at the novelty of the present work.
Materials and Methods:
The materials and methods section lacks the necessary detail to ensure reproducibility. Please provide comprehensive descriptions of the procedures used for soil analysis. As the plants were planted back in 2015, authors must provide their nutritional history. Experimental design still need clarifications. Please clarify that if 2m x 1.5m is size of main plot or sub-plot? If 400 seeds were applied per square meter it means in 1 plot of 3 m2 size authors seeded 1200 seeds? that quite a big number. Please check it again. Clearly justify all the information about sub-plot experimental site, non-experimental sites etc. to justify that it was a split-plot experiment.
Results:
Results are okay, however authors are advised to draw a correlation plot for nutrients in switchgrass cultivars, at different harvest intervals and years with soil physicochemical properties to trace the influence of soil nutritional properties on the plant parts.
Discussion:
Strengthen the discussion by integrating your findings with the latest references. Provide a thorough explanation of how your results compare with previous studies and what new insights they offer. Relate the nutrient profile of plants with soil pH and Ec and other important factors. Address any limitations of your study, such as environmental conditions, potential confounding factors, or methodological constraints. Discuss how these limitations might affect the interpretation of your results. Highlight future research directions and practical applications of your findings.
Figures, Tables and Supplementary Material:
Figure presentation needs further refinement and adjust them to 600 DPI. In Figure 1, 2 and 3 only add letters to show the level of significance and remove the means. In footnotes of the figures add what does these letters on bars are indicating. Refine the captions of the figures that briefly explain what is depicted inside the figures. Divide the sub-figures into different parts by assigning letter a, b, c...and so on and then explain e.g. Fig 1a) phosphorus content. In tables captions explain all the abbreviations and parameters that are presented inside the table. In Table 2 please add standard deviations and letters indicating level of significance. Please provide supplemental data related to soil analysis (replicates readings etc.)
Conclusion:
Conclusion must be generalised and evidence based and must not be a repetition of the results. Please add future recommendations in the conclusion section.
References:
Please double-check the references to ensure that all references cited in text are present in reference list. Please remove the outdated and obsolete references.
Due to these significant issues, the quality of the manuscript currently does not meet the standards of PeerJ and requires major revisions.
Specific comments
1)In lines 44-47, the authors suddenly discuss animal diseases, which is confusing. It gives the impression that the authors are working on this particular disease: “The dry matter’s Ca/P ratio, regardless of cultivar or harvest stage, posed no risk to animal health. The K/(Ca+Mg) ratio varied between 0.807 and 1.235 depending on the cultivar and harvest stage, and these ratios do not pose a risk of tetany illness in terms of animal health.”
2)Line No. 36-37, clarify this statement why you used this specific term “The Long Island”, what is its link with your experiment.
3)Line No. 38, please add unit after values (% 0.223).
4)Line No. 58-59 please keep most relevant and latest two references, and avoid to cite more than three references in a single turn, please make such type of corrections throughout the manuscript “(Sanderson et al., 2006; Liebig et al., 2008; Robertson et al., 2011; Balsamo et al., 2015).
5)Line No. 59-61 Please provide solid background information regarding its use in silage and discuss its economic importance as a fodder. “Switchgrass can be utilised as forage for ruminant animals through grazing, as well as for silage and hay as roughage (Moore et al., 2004; Zhao et al., 2017; Taranenko et al., 2019; Eli_ and Özyaz1c1, 2019).”
6)Please provide reference for this statement and correct its expression in Line 71-73 “Animals’ mineral needs vary based on species, breed, age, gender, growth, health, pregnancy, and milk output, as well as the quantity and bioavailability of minerals consumed”.
7)Please discuss the topic within a broader context and its economic influence, avoiding irrelevant information as given in Line No. 76-78 “Loss of appetite, decreased productivity, cachexia, hair loss, changes in skin and hair color and structure, tooth breakage, abortions, anorexia, diarrhea, anemia, digestive system and bone disorders, and diseases like pica can occur.
8)In Line No. 116-118, please correctly include information regarding the main plots and replicates in the experimental design section rather than in analyses section. “The study consisted of 32 plots in total, with 8 varieties and 4 replications, and samples were taken from the same plots each year. From the green forage harvested from each plot, 500-gram samples were randomly taken.
9)Please provide standard experimental references instead of citing anonymous sources, as mentioned in lines 119-120. “Total P, K, Ca, and Mg concentrations in the ground hay samples were determined using an IC-0904FE calibration set (Anonymous, 2020b)”
10)In lines 212-215, please clarify and draw a relationship between current meteorological data, plant minerals, and soil status. This is crucial for the authenticity and clarification of the obtained results, as the authors mentioned that the soil pH was 7.95 in 2019 and 7.98 in 2020. Since phosphorus is usually immobilised at a basic pH, it hinders plant uptake. Even excess precipitation can also cause mineral leaching from the soil. “According to Roche et al. (2009), plant uptake of mineral compounds depends on various parameters, and meteorological conditions such as temperature, duration of sunlight, and precipitation, notably soil and air temperature, influence the concentration of mineral substances in plants.”
11)Please avoid citing obsolete references, such as those given in line 244 (Tejeda et al., 1985, and Kidambi, Matches & Griggs, 1989). Remove these outdated references and instead consult and cite more recent research, preferably not older than 2000. please ensure to make such corrections throughout the manuscript.
12)Please discuss your obtained results logically and avoid linking them to animal diseases. as in Line No. 263-266 “Grass tetany is a disorder that develops slowly in calves and beef cattle, and rapidly in dairy cattle............”
13)Please italicise all scientific names as in Line 290 Dactylis glomerata L.

Experimental design

Experimental design seems okay but write up needs further clarification to ensure consistency and coherence.

Validity of the findings

The findings are promising. However, the authors have to do rigorous work to draw an exact relation between soil physico-chemical characteristics, plant nutrient level, cultivars, years and climatic data.

Additional comments

The manuscript can provide some applicable knowledge to scientific community in terms of forage, roughage nutrition for animal if particular switcgrass spcies are used.

Reviewer 2 ·

Basic reporting

Although the article represents a significant effort to evaluate the nutritional status of switchgrass at different growth stages, there are several notable shortcomings in its presentation:
1-The article lacks a comprehensive description of the methodology about sampling technique, and statistical methods are either used incorrectly or insufficiently explained. This makes it difficult to evaluate reliability of the results.
2-There is a noticeable absence of comparative analysis with other forage crops (or very old studies e.g from 1985) or with switchgrass harvested at different locations. Including such comparisons would provide a more comprehensive view of how switchgrass performs relative to other species or under varying conditions.
It also needs following corrections:
Line 72: Run-on sentence. Restructure/rephrase to make it clear.
Line 95-97: Remove full stop after ‘Province’, and write ‘Table’ with small t
Line 117: Wrong Spellings replace ‘Analyzes’ with analyses
Line 295 italicize Dactylis glomerata
Line 314-15: Not the preview of the study and hence should not be included in the manuscript.

Over all it needs lot of language clean-up, better sentence composition and avoiding run-on sentences.

Experimental design

The alpha values for poorly controlled (or discussed) for pairwise comparison test as discussed, and every time it was used 0.05. If Tukey’s Test was used then it controls error FWER.
Tukey's HSD test is relatively conservative, meaning it tends to be more stringent in declaring differences between group means as statistically significant. This can result in Type II errors (failing to detect a difference when one exists), especially in studies with small sample sizes or when differences between means are subtle.

Although Tukey's HSD controls for the family-wise error rate (the probability of making one or more Type I errors when performing multiple comparisons), it does so by increasing the likelihood of Type II errors. This balance between Type I and Type II errors can be a disadvantage when researchers are interested in detecting as many differences as possible.

Therefore, Duncan Multiple Range test is suggested due to following two reasons:

a) DMRT is generally considered less conservative than Tukey’s HSD, which means it can be more powerful in detecting differences between group means. This can be advantageous when the goal is to identify subtle differences.
b) DMRT allows for multiple comparison procedures and can be adjusted for different significance levels, providing some flexibility in analysis.

I believe you should re-do the statistical analyses, by modelling the effect of year, and compare the means using DMRT. It will help you draw better inferences about the cultivars and harvest stage, will make tables less cluttery. It will also make tables more meaningful, and will have better understandings of the study.

Validity of the findings

Tables present lots of data which is meaningless and not discussed in the manuscript.
Table title very poor and brief, does not qualify for the criteria of “Stand alone status”
Moreover, lettering should be superscripted
In Table 4, Mg analysis does not have letters indicating differences for cultivar x harvest stage values in last two columns.

Table 3: Year has significant effect on all the nutrients (except P) studied in the manuscript. The significant effect is confounded between the biological status of the grass and the environmental condition of that particular year. It is unclear whether as the matter of fact Grasses tended to increase the harvest yield and decrease in quality as they grow older OR change in temp and humidity is making the impact significant (as first year of the trial has higher total precipitation compared to second year).

Reviewer 3 ·

Basic reporting

The manuscript presents the results of a two-year field experiment in which the effect of cultivar and harvest date on the content of selected macroelements in the biomass of switchgrass was analyzed. The field experiment was conducted correctly. The obtained results were interpreted based on statistical analysis. Tables and Figures are well prepared, understandable and legible.

General note on the manuscript: It is a pity that the authors focused only on the content of these 4 macroelements. The authors omitted completely different parameters, which to a greater extent determine the nutritional value of green fodder or switchgrass hay, i.e. dry matter content, protein content, fiber content and its fractions. Presentation of these features would significantly increase the scientific value of the presented research
Detailed notes:
1. Keywords used when applying the manuscript
2. The 'Introduction' chapter is quite general and needs improvement. In this part of the work, the existing state of knowledge regarding the nutritional value of switchgrass must be presented. What is known and what still needs to be explained. The results of studies that have already been conducted must be presented in more detail. How does the harvest date affect the nutritional value of switchgrass, as well as other perennial grasses.
3. Chapter 'Material and Methods'
- The manuscript provides soil properties for the experimental field. Please describe how soil samples were taken, how many, when, and what methods were used for analysis.
- The description of weather conditions is too general. The data in Table 1 show that in some months the differences between 2019 and 2020 were significant.
- Please provide dates for individual harvests in 2019 and 2020.
- Please explain how irrigation needs were determined. How much water was used in each year.
- in line [109] is that “Accordingly, the cultivars were main plots and harvest stages were subplots, forming a split-plot design’ but in line [128] is “All data obtained from the study were subjected to analysis of variance (ANOVA) according to a randomized complete block design”.
4. The results should also include the yield of switchgrass biomass. I believe the authors have determined the yield of harvested biomass.

Experimental design

no comment

Validity of the findings

no comment

Additional comments

no comment

Reviewer 4 ·

Basic reporting

The manuscript by Acikbas examines the effect of harvest time on forage nutrient content in eight cultivars of switchgrass. The manuscript found that nutrient content varied considerably among cultivars and harvest times. It also found that while switchgrass provided insufficient nutrition as a primary livestock feed, it could be used to supplement livestock feed with little-to-no risk to livestock health.

It would be useful to readers if the raw data file contained descriptive meta-data.
Overall, the manuscript is well written. I point out a few places below where language could be clarified.
Lines 75-78: Can this list of symptoms be condensed? It seems unnecessarily long here.
Lines 93-94: This isn’t a complete sentence -- reword
Line 159-181: It seems like ‘ratio’ and ‘content’ are being used interchangeably here. To limit confusion, I’d suggest calling the percent of an element in samples ‘content’ (e.g., ‘P content’) and the relationship between two elemental contents ‘ratio’ (e.g., ‘Ca/P ratio’).
Line 133 and elsewhere: It would be better to call this ‘analysis of variance’ rather than ‘variance analysis’.
Lines 134-149: Here and elsewhere, it would be valuable to list the P values rather than just declaring them significant/non-significant.
Line 255: Isn’t a ratio of 2:1 the same as a ratio of 1:0.5?
Figs 1-3: I would recommend removing the means from the top of the error bars in these figures – they can be estimated by readers from the figure itself and, more importantly, distract somewhat from the compact letter display, which is needed for readers to assess whether groups differ significantly from one another.

Experimental design

Lines 93-98: It’s not clear how far the station measuring these meteorological data is from the field site and how similar the meteorological site and the field site are. Can that be added?
Lines 100-102: This description is a little unclear and would benefit from editing. First, what was the consequence of the plot spacing? Was these fairly narrow rows lead to closed canopy plots? Did the plant crowns interact with one another? Did each 2m x 1.5m plot include 10 rows of plants of the same cultivar?
Lines 107-108: It would be helpful to readers to describe these harvest stages in more detail. Additionally, how consistent was phenology across plants within each plot?
Lines 109-110: So each plot was split into three subplots? Were the subplots randomly assigned to each harvest stage?
Line 117: In switchgrass, plants often start to senesce following flowering. Did the proportion of green tissue change between harvest stages? If so, was this accounted for in sampling to maintain consistency across harvest stages?
Lines 211-212: I’m curious whether there were any differences in the date that plants were harvested for each harvest stage between the two years. Could something like that explain a portion of this growing season effect?

Validity of the findings

Lines 127-128: If the experiment was a split-plot (as described at lines 109-110), then treating the analysis as a randomized complete block design is inappropriate. The cultivar, which is the whole plot effect, should have fewer denominator degrees of freedom than harvest stage, which is a subplot effect. I think these analysis of variance models should be refit with a subplot-level effect. Some of the P-values of the whole-plot effect (cultivar) may become non-significant in an appropriately parameterized model, but the subplot effect (harvest stage) and interactions with harvest stage will probably not change much (if at all).

---

## Round 0.2 · Minor Revisions

Please revise manuscript as per the remaining advice from the reviewers.

·

Basic reporting

The authors have addressed most of the concerns raised, but the following issues must be resolved to further improve the quality of the manuscript and make it suitable for publication in the journal. There are still some typographical errors and instances of awkward phrasing. Please thoroughly review the manuscript for these issues and correct them accordingly.
Please include a correlation analysis to examine nutrient variations in switchgrass cultivars across different developmental stages. A single correlation plot would not significantly increase the length of the paper but would provide a concise and clear representation of the relationships between the wide range of parameters measured in the experiment.
Refine the captions of the figures to briefly explain what is depicted in each sub-figure. There is no need to reorganize the figures; simply assign alphabetical labels (e.g., Fig. 1a, 1b, etc.) to each sub-figure and describe them in the caption, such as "Fig. 1a) Phosphorus content," ensuring clarity and easy reference.
In discussion, the references to previous studies are useful, but they are largely descriptive. The authors should engage more critically with the literature to show how their findings align or contrast with existing knowledge. Furthermore, soil pH plays a critical role in influencing the availability of phosphorus and other essential minerals. It is recommended that the authors incorporate a discussion on the effects of soil pH and electrical conductivity (EC) on mineral availability, as these factors significantly impact nutrient uptake and overall mineral balance in plants.

In conclusion section, while the practical recommendation for feed supplementation is useful, the conclusion would benefit from addressing future research directions. For instance, would breeding efforts or soil amendments help improve P and Ca uptake in switchgrass? Additionally, could different management practices (e.g., adjusting harvest times) optimize nutrient retention in the plants? These suggestions would elevate the conclusion from a summary to a forward-looking piece that offers solutions beyond short-term supplementation.

The final sentence in conclusion is useful but could be rephrased for stronger impact. Instead of stating that "supplementation... may be necessary," a more definitive statement like "supplementation... is recommended to ensure optimal productivity" would give clearer guidance to stakeholders.

Experimental design

The experimental design is satisfactory

Validity of the findings

The validity of the findings could be strengthened if the authors establish a relation between the nutrient content in plants at different growth stages and the availability of nutrients in the soil, considering factors such as soil pH, electrical conductivity (EC), and organic matter content.

Reviewer 2 ·

Basic reporting

The authors have made considerable changes in the article and the manuscript has improved significantly.

However, a few concerns are still to be addressed. For example, It was suggested to avoid run-on sentences, and one sentence from the intro was quoted as an example. The authors corrected that sentence only. Using short sentences with a single focus enhances article clarity and sharpens readability.

Experimental design

It was suggested:
There is a noticeable absence of comparative analysis with other forage crops (or very old studies e.g from 1985) or with switchgrass harvested at different locations. Including such comparisons would provide a more comprehensive view of how switchgrass performs relative to other species or under varying conditions.

The authors wrote a perplexed answer to the comments. It is suggested to bolster the article with recent studies. There are two options for that:
1- Find the studies in the same grass for micronutrients studies
2- if there are no studies found, examples should be quoted from closely related species.

Validity of the findings

Looks good

Reviewer 3 ·

Basic reporting

The authors have taken into account some of the comments from the review in the revised manuscript. However, I still believe that additional information is needed
1. Please describe what methods were used for soil analysis. It is not about describing in detail the methodology of these analyses. Please state what methods were used. Even if we order analyses, the contractor states what methods he used to make the analyses.
2. Please explain how irrigation needs were determined. How much water was used in each year. In my opinion, this information is not sufficient “The switchgrass experimental field was properly irrigated each year using a drip irrigation system”
3. The author's explanation for not performing other analyses that would determine the nutritional value of switchgrass biomass can be accepted. However, when deciding on fragmentary studies, one must be aware that they may not be interesting enough to be accepted for publication in recognized international journals. I do not understand the explanation that for economic reasons the author did not mark the biomass yield. After all, it does not cost anything. You just have to spend some time. After all, switchgrass was harvested.

Experimental design

It is correct

Validity of the findings

The content of macroelements in the feed is important, although it is a pity that other properties that are more decisive for the nutritional value of switchgrass biomass were omitted.

Reviewer 4 ·

Basic reporting

The revised manuscript by Acikbas is much improved and has addressed most of my previous comments. I mention a few points of clarification below.

Supplementary Table 5: Metadata describe the data presented in the supplementary file. For instance, without reading the manuscript, it’s not obvious which columns are percentages and which are ratios. Similarly, it’s unclear without reading the manuscript what the two ratio column headings mean (‘Ca-P’ and ‘K (Ca+Mg)’) and what year = 1 and year = 2 are. Instead of requiring future users to read through the entire paper to glean this information, it’s valuable to include it in the raw data file. I still think this would be a valuable addition to the manuscript.

Line 61: Better to say ‘native to North America’, because switchgrass also occurs in Mexico and Canada.

Line 287: Should this be “Ca content” instead of “Ca ratio”?

Experimental design

Lines 132-133: If I understand this, the uplands had an earlier phenology and were, thus, cut earlier than the lowlands. If that’s the case, I’d reword this as ‘reached the cutting stages earlier’ and ‘reached the cutting stages later’, respectively.

Lines 134-141: Specify somewhere in this section that the dates (e.g., ‘July 8-21 and July 10-19’) represent 2019 and 2020, respectively.

Validity of the findings

I appreciate that the revised manuscript takes a more appropriate statistical approach than the original manuscript.

---

## Round 0.3 · accepted · Accept

The manuscript is accepted after satisfactory revision.